# Associating Objects and Their Effects in Video through Coordination Games

**Erika Lu**[1]         **Forrester Cole**[1]         **Weidi Xie**[2]         **Tali Dekel**[1,3]

**William T. Freeman**[1]         **Andrew Zisserman**[4]         **Michael Rubinstein**[1]

[1] Google Research   [2] Shanghai Jiao Tong University   [3] Weizmann Institute   [4] University of Oxford

## Abstract

We explore a feed-forward approach for decomposing a video into layers, where each layer contains an object of interest along with its associated shadows, reflections, and other visual effects. This problem is challenging since associated effects vary widely with the 3D geometry and lighting conditions in the scene, and ground-truth labels for visual effects are difficult (and in some cases impractical) to collect. We take a self-supervised approach and train a neural network to produce a foreground image and alpha matte from a rough object segmentation mask under a reconstruction and sparsity loss. Under reconstruction loss, the layer decomposition problem is underdetermined: many combinations of layers may reconstruct the input video. Inspired by the game theory concept of focal points—or *Schelling points*—we pose the problem as a coordination game, where each player (network) predicts the effects for a single object without knowledge of the other players' choices. The players learn to converge on the "natural" layer decomposition in order to maximize the likelihood of their choices aligning with the other players'. We train the network to play this game with itself, and show how to design the rules of this game so that the focal point lies at the correct layer decomposition. We demonstrate feed-forward results on a challenging synthetic dataset, then show that pretraining on this dataset significantly reduces optimization time for real videos.

## 1   Introduction

Identifying and associating effects such as shadows and reflections with the objects that produce them is a fundamental and difficult task for visual understanding. A variety of cues such as motion, appearance, and proximity may be used to match an object and an effect, but these cues may vary widely depending on the 3D geometry and lighting conditions of the scene, and nearby objects may produce similar effects (Fig 1). There is no simple heuristic to reliably connect an object and the visual effects it generates.

The recent *Omnimatte* approach [11, 12] learns to associate an object with its effects by training a CNN to decompose the video into layers, where each layer contains the object as well as the effects associated with that object. The CNN is given a binary segmentation mask of an object, omitting any associated effects, and learns to output a foreground color and alpha matte for that object and its effects. Notably, no explicit loss is provided for the association of objects and effects: the correct association emerges through the structure of the optimization and the inductive bias of the CNN. The optimization requires hours of processing for each video, however, and is vulnerable to errors when multiple objects have closely correlated motion (Fig 5).

36th Conference on Neural Information Processing Systems (NeurIPS 2022).

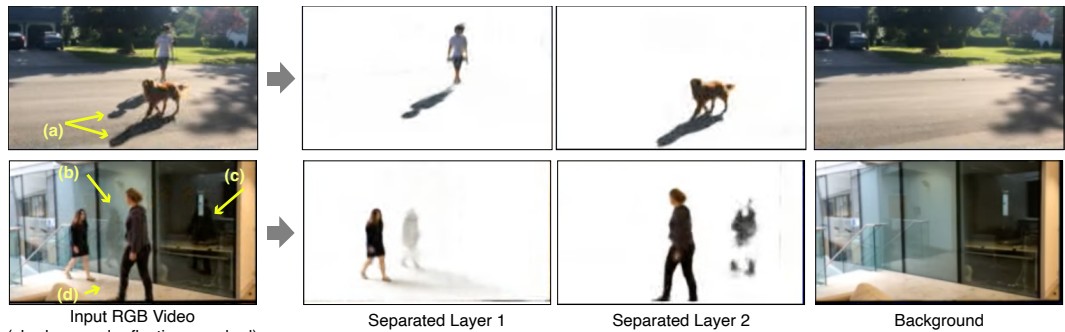

| Input RGB Video (shadows and reflections marked) | Separated Layer 1 | Separated Layer 2 | Background |

Figure 1: Example videos and our decomposition result with shadows and reflections. Matching objects and their effects is challenging: the true object may be further from its shadow (a) or reflection (b) than another object, and effects may be subtle (reflection (c), soft shadow (d)).

This paper aims to solve the layer decomposition problem directly by training a feed-forward neural network to predict the layers from the input video and masks. Since ground-truth labels for effects are not available for real videos, we take a self-supervised approach and train the network only with a reconstruction and sparsity loss. Unfortunately, reconstruction and sparsity losses do not uniquely determine a layer decomposition: effects may be shuffled between layers while producing the same loss (Fig 3). For example, a valid but incorrect solution puts the entire video in the front layer, leaving the remaining layers blank.

To solve this problem, we take inspiration from the game theory concept of focal points, also known as Schelling points [16]. In a coordination game, the players must cooperate to achieve a goal without communicating with each other ahead of time. A classic example is giving several people the instruction: "Meet in New York City on Tuesday", without giving any details about the exact location and time when they should meet. Schelling found that the most common strategy, based on the players' own prior knowledge of the city, was to go to Grand Central station at noon of that day. Players converge on such a "natural" solution, or focal point, in order to maximize the likelihood of their choices aligning with those of other players. In our case, the network plays a coordination game against itself: it is given an object mask and is asked to predict the appearance of that object and its visual effects, without knowledge of the other network instances' choices. We show that by training the network to predict masked video frames, we can engineer the focal point of this game to lie at the correct layer decomposition.

We demonstrate the success of the approach on a challenging synthetic dataset of moving objects with realistic shadows and reflections. The learned prior helps the network overcome ambiguous inputs, leading to improved quality over single-video optimization [12]. The prior learned from synthetic data also provides a good starting point for optimization on real videos. Fine-tuning the pretrained network on a new video reduces optimization time by $10\times$ over single-video optimization, while producing similar layer decomposition quality.

## 2   Related Work

**Video layer decomposition.** Separating videos into layers is a fundamental computer vision problem that has been studied for decades [18]. It is important for both basic understanding of scenes (e.g., for estimating depth [24] and occlusion boundaries [2]), as well as for supporting various operations on videos such as editing (e.g., [1]) and view synthesis (e.g., [17]). Recently, several works besides the previously mentioned Omnimatte [11, 12] have applied deep learning to make progress on this problem. Zhang et al. [23] generate editable free-viewpoint videos of dynamic scenes from sequences captured by multiple cameras through a learned, layered neural representation that disentangles location, deformation, and appearance of dynamic objects. Kasten et al. [9] propose a decomposition method that represents a video as a set of layered 2D atlases that are global to the video, along with associated (per-frame) alpha maps. Like our work, these methods are self-supervised. However, they involve slow optimizations, overfitting neural networks' weights to an input video. In contrast, this work aims to solve the decomposition problem directly by training a feed-forward model to predict the layers from the input video and masks.

**Specialized methods and supervised approaches.** Several methods exist that learn to decompose videos or detect objects' effects (such as reflections and shadows) in a supervised manner. "Visual Centrifuge" [1] separates videos with transparency and reflections into layers, by training a neural network on a large and diverse set of synthetically blended videos. This method learns to separate difficult effects like shadows and reflections, but does not address the problem of associating those effects with the objects that produce them. Other works such as [10, 7, 22] train a network to detect and remove shadows specifically by generating synthetic shadow/no-shadow image pairs and using them as training data. These methods focus on directly removing shadows, not decomposing the video into layers as in our approach. Like our approach, instance shadow detection [20, 19] aims to associate objects and their shadows. They take a supervised approach, however, and manually label a large dataset of photographs. Hu, et al. [6] collect an even larger dataset of manually labeled shadows. Manual labeling is only practical for hard, clearly visible shadows, however, so models trained on these datasets are limited to those effects. In contrast, our method makes no assumptions about the appearance of shadows, reflections, or other effects, and requires no data labeling step.

**Structured representation for images and video.** Our approach is most similar to recent methods for image and video understanding that aim to produce an interpretable, structured representation through self-supervised learning. Methods such as [8, 13, 4] learn keypoint-based representations for predicting future video frames by training a decoder to map keypoints to RGB through a pixel reconstruction loss. The learned keypoints are useful for a variety of downstream applications, such as trajectory prediction and action recognition. Like these methods, we learn a structured representation using future frame prediction, but where keypoint representations have fewer variables than the original video, our layer decomposition has *more* variables, making the optimization problem underdetermined. Image de-rendering methods [21, 3] aim to construct a detailed scenegraph description of an image that can be both re-rendered using conventional computer graphics and used for alternative tasks like image captioning. Like these methods, our reconstruction stage is a fixed-function operation (alpha compositing), but we operate on video, and our representation is specifically targeted at matting and associating objects with their visual effects.

# 3 Method

The goal in this paper is to train a feed-forward network that can decompose an RGB video into *omnimattes*, layers that capture an object and all the scene effects related to that object. We give a formal definition of this problem in Section 3.1. We choose a network architecture that combines CNNs and a transformer decoder, which allows capturing long-range effects, and binding the objects with their effects (Section 3.2). The proposed architecture is trained purely with self-supervised learning (Section 3.3), which enables further improvement of the results via test-time training on real video sequences (Section 3.4).

## 3.1 Problem Definition

For a given video with $T$ frames, we aim to train a feed-forward neural network that decompose the frame containing $N$ dynamic objects into *omnimattes*; without loss of generality, we present the problem definition based on frame $t$:

$$\{\mathcal{L}_t^1, \ldots, \mathcal{L}_t^N\} = \Phi(I_1, \ldots, I_T, M_t^1, \ldots, M_t^N; \Theta) \tag{1}$$

where $\{I_1, \ldots, I_T\}$ denotes the input video sequence, $\{M_t^1, \ldots, M_t^N\}$ denotes $N$ binary segmentation masks, each corresponding to one object of interest, similar to [12]. $\mathcal{L}_t^i$ refers to the output $i$-th RGBA (color and opacity) layer, representing one of the objects and all the scene effects that are related to it.

We assume known (possibly time-varying) compositing order $o_t$, and known background $\mathcal{L}_t^0$, which can be either given by the synthetic training data or obtained using a simple background estimation process in the case of a static background (e.g. median pixel value). The input images $I_t$ may be reconstructed by compositing $\mathcal{L}_t^i$ using standard alpha compositing [15], $I_t \approx \text{Comp}(\mathcal{L}_t, o_t)$.

Unfortunately, the layer decomposition problem is ill-defined, with multiple sets of $\mathcal{L}_t^i$ that reconstruct $I_t$. For example, one obvious solution would be to place all objects and effects in one layer and leave the others empty, *i.e.* $\mathcal{L}_t^i = I_t$, $\mathcal{L}_t^j = 0$, $\forall j \in [1, N]$, $j \neq i$. Thus, in order to train a feed-forward network for associating objects and effects, the model must be trained to exploit the correlations in

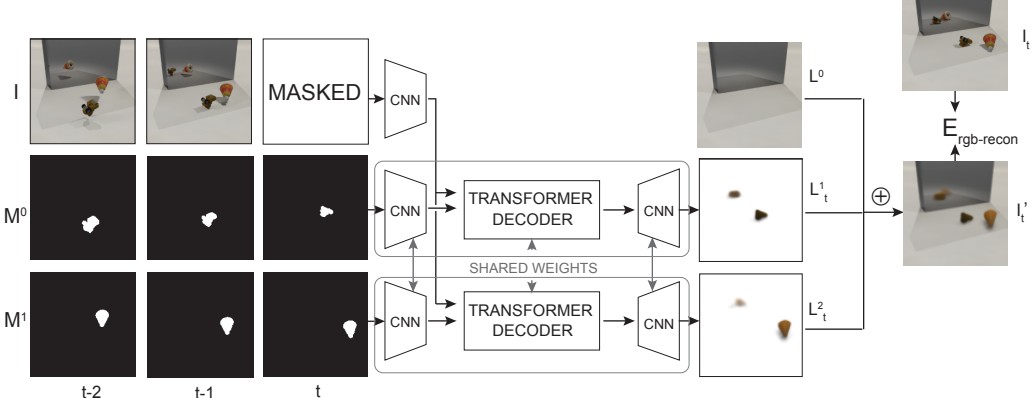

Figure 2: Overview of the training pipeline. The input is a short RGB video clip $I$ and masks $M^i$. The target frame $I_t$ is masked from the input of the network. The transformer decoder is given features extracted from the RGB input and a single mask layer, and produces a single output layer. The output layers $L_t^i$ are composited over the background $L_0$ to form the predicted frame $I_t'$, which is compared to the target frame $I_t$. No direct supervision is provided for $L_t^i$.

video data and avoid learning trivial solutions. Our solution is to run the network separately on each layer, and task it with predicting unseen frames of the video from object masks and nearby frames. We give intuition for this approach as a coordination game with a designed focal point in Section 3.5.

## 3.2 Architecture

Our model combines CNNs and Transformer modules; an overview is shown in Figure 2. For an input video sequence with $T$ frames, we parameterize the image encoder with a shared CNN, $\mathcal{F} = \{f_1, \ldots, f_T\} = \{\Phi_{\text{enc}}(I_1), \ldots, \Phi_{\text{enc}}(I_T)\}$, where $I_i \in \mathbb{R}^{H \times W \times 3}$, and $f_i \in \mathbb{R}^{\frac{H}{16} \times \frac{W}{16} \times D}$, where $H$, $W$, $D$ are the spatial height, width and channel dimensions. Similarly, the objects in frame $t$ are represented by their *rough* binary segmentation mask, and fed to another CNN, $\mathcal{M} = \{m_1, \ldots, m_N\} = \{\Phi_{\text{mask}}(M_t^i), \ldots, \Phi_{\text{mask}}(M_t^N)\}$, with each $m_i \in \mathbb{R}^{\frac{H}{16} \times \frac{W}{16} \times D}$.

As input to the transformer decoder, we treat the image features as keys and values, with the positional information injected by learnable spatio-temporal embeddings. The encoded masks are treated as queries to the transformer decoder independently, *i.e.* only one of the object's masks are used as the query at a time. The output query maintains its spatial resolution, and we further upsample the features with a CNN decoder to produce the final RGBA layers corresponding to the input masks.

**Discussion.** Here, we have made two critical choices on the architecture design: first, we use a transformer decoder for associating objects with their effects, this architecture is naturally suited to our task, as self-attention allows the queries to learn across multiple frames, to determine object motion and deformation, and thus identify the effects that are correlated with the object of interest in motion and/or shape; Second, each object's layer is predicted without knowledge of the other objects, which enables to correctly assign scene effects to the layer of the object that causes them. Additionally, running the network separately on each layer also has the benefit of generalizing to an arbitrary number of layers.

## 3.3 Self-supervised Learning

To train the proposed architecture, we define the proxy task as frame reconstruction. Specifically, we randomly mask a subset of frames from the input sequence, and task the model to only reconstruct the unseen frames based on the input object masks and other frames in the sequence.

In detail, we follow the same loss formulations as [12], with minor modifications. The primary loss for training is a reconstruction loss $\mathbf{E}_{\text{rgb-recon}}$, with sparsity regularization losses $\mathbf{E}_{\text{reg}}$ to encourage sparse alpha mattes, and a mask initialization loss $\mathbf{E}_{\text{mask}}$ to bootstrap training. The reconstruction loss is defined as:

$$\mathbf{E}_{\text{rgb-recon}} = \frac{1}{T} \sum_t \|W_t \odot (I_t - \text{Comp}(\mathcal{L}_t, o_t))\|_1, \tag{2}$$

where $\odot$ is element-wise product and $W_t = 1 - \lambda_\text{w} \sum_i M_t^i$ is a spatial weight to emphasize reconstruction of shadows and reflection effects over object regions.

The alpha sparsity term, aimed to prevent all layers from trivially reconstructing the entire frame, is:

$$\mathbf{E}_\text{reg} = \frac{1}{T}\sum_t \gamma \left\|\text{Comp}(\alpha_t, o_t)\right\|_1 + \Phi_0(\text{Comp}(\alpha_t, o_t)), \tag{3}$$

where $\Phi_0(x) = 2 \cdot \texttt{Sigmoid}(5x) - 1$ smoothly penalizes non-zero values of the alpha map, and $\gamma$ controls the relative weight between the terms.

The bootstrapping, or mask initialization, loss is:

$$\mathbf{E}_\text{mask} = \frac{1}{T}\frac{1}{N}\sum_t\sum_i \left\|b_t^i \odot (M_t^i - \alpha_t^i)\right\|_2 \tag{4}$$

where $b_t^i$ is a spatial weight that balances the loss in positive and negative mask regions. This bootstrapping loss ensures the optimization starts from a roughly accurate layer decomposition, where the predicted alpha maps $\alpha_t^i$ match the input masks $M_t^i$.

The complete loss term is:

$$\mathbf{E}_\text{rgb-recon} + \lambda_\text{r}\mathbf{E}_\text{reg} + \lambda_\text{m}\mathbf{E}_\text{mask} \tag{5}$$

where $\lambda_\text{r}$ and $\lambda_\text{m}$ are weighting coefficients. The weight $\lambda_\text{m}$ is gradually decreased during training to reduce the effect of the bootstrapping loss as the optimization converges (see supplemental material).

### 3.4 Training Procedure

Training proceeds in two stages, namely, pre-training and test-time training.

**Pretraining.** At this stage, we pre-train our model on a large corpus of video data, specifically, we sample a sequence of $T$ video frames and their corresponding object masks, we then select a subset of $K$ frames to mask, or hide from the network, and tasked the model to reconstruct these the unseen frames based on their input object masks.

**Test-time finetuning.** As our proposed model is trained from purely self-supervised learning, it is possible to adapt to any test video sequence in the same form of optimisation-based approach, supporting the model to generalise to a different data domain, e.g. sim2real. For this test-time training stage, we use the same losses and training setup as for pretraining, with the exception of a lower weight for the bootstrapping loss.

### 3.5 Layer decomposition as a coordination game

The reconstruction task can be seen as a co-ordination game between multiple players: $N$ instances of the network $\Phi_\Theta$. Each is responsible for producing a single layer $\mathcal{L}_t^i$, and each instance is blind to the layer produced by the others. The (negative) total loss Eq. 5 is the reward shared by all players. Even with the sparsity regularization term $\mathbf{E}_\text{reg}$ and mask term $\mathbf{E}_\text{mask}$, the reconstruction is underdetermined: for any given decomposition $\mathcal{L}_t^i$, an equally good reconstruction and sparsity score is obtained by removing effects from one layer and adding them to another (Fig. 3). Any decomposition that minimizes Eq. 5 may be thought of as a Nash equilibrium [14] of this game.

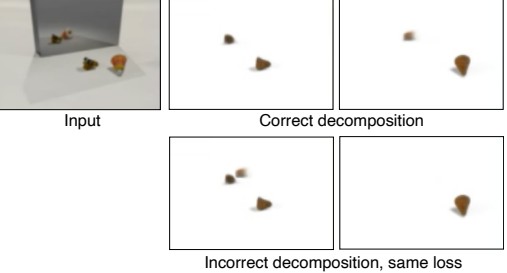

Figure 3: Ambiguity of the decomposition problem. Under a reconstruction loss, effects may be moved between layers without cost. The task and architecture must be designed to produce the correct (top) instead of an incorrect (bottom) solution.

In a coordination game with multiple equilibria, the players tend to choose the most *prominent* equilibrium, otherwise know as a *focal point* or *Schelling point*, based on some shared knowledge the players bring from outside the game [16]. Importantly for our case, the most prominent equilibrium need not be explicitly defined (i.e., supervised). Instead, it may be implicitly defined by the shared knowledge of the players. Over many rounds of this game, the players learn rules to find a focal point, without the rules or the focal point

being explicitly specified. In other words, this approach allows the players (multiple instances of the network $\Phi_\Theta$) to be trained with self-supervision. The exact focal point that is found through this method depends on the structure of the task. By asking each instance to produce a single layer, providing a single object mask, and bootstrapping the network to match that mask, we bias the network towards explaining a single object with each instance. As shown by Lu, et al. [11], effects that are correlated with the object mask are easier to predict than uncorrelated effects, so a single object's associated effects tend to be predicted in the object's layer. Forcing the network to predict the layers of an unseen frame, rather than providing the entire video clip as input, helps train the network to find and exploit these correlations (Section 5.4).

# 4 Dataset Description

For our experiments, we use a mixture of synthetic and real data. Specifically, we use the Kubric codebase [5] to generate a large-scale synthetic data for pretraining, while our model is self-supervised, and can thus be pretrained on real data, we choose to use synthetic data for this stage due to the costly procedure for curating video datasets with interesting effects, such as shadows and reflections, and which also meet our constraints (a background that can be well-estimated, and objects undergoing significant motion). An additional benefit of a synthetic dataset is that the objects can be rendered individually, providing a ground truth decomposition that can be used for numerical evaluation, so long as the original scene does not contain collisions between objects.

## 4.1 Synthetic shadows and reflections

For all synthetic videos, we use a randomly selected high dynamic range image (HDRI) as the background texture. Objects are randomly sampled from the Google Scanned Objects (GSO) dataset of 3D scanned household items. We split the HDRIs into 450 train and 59 test images. For the GSO objects, we use Kubric's default train/test split (90/10) of the 1033 objects.

We render all videos under a fixed camera position with three light sources at random positions in the scene, and an additional 4th fill light at a fixed location. We use default settings to initialize the position and velocity of objects. Each video is 2s long at 12 fps, totalling 24 frames.

**Shadow dataset** is generated to understand our model's ability to associate objects with their shadows. For training, we render 10k videos with 2 objects, while for numerical evaluation, we only keep the generated test videos that do not contain object collisions, resulting in 323 2-object test videos and 86 4-object test videos, with the latter to test our model's ability to generalize to more objects than seen during training.

**Reflection dataset** is generated with under the same physical conditions as the Shadows dataset, but with an additional mirror object that reflects about 50% of the scene.[1] We randomly jitter the placement and orientation of the mirror. As with the Shadows dataset, we generate 10k 2-object training videos, and 315 2-object videos and 38 4-object videos with objects not colliding.

## 4.2 Real videos

Apart from evaluating on the synthetic data, we also test on the challenging real sequences with various effects. We use real videos from Lu et. al [12] for comparison, including `Reflections`, which contains two people passing each other with their reflections visible; `Trampoline`, which involves six people jumping on trampolines; `Dogwalk`, which involves a person walking a dog while both cast long shadows; and `Soccer`, which contains a player kicking a ball and also casting long shadows. These videos range in length from 80 - 200 frames, sampled at 30 fps.

---

[1] At the time of submission, Kubric did not support reflections for HDRI backgrounds; thus the mirror contains the reflections of the scene objects but not that of the textured background.

# 5 Experiments

## 5.1 Training details

For our experiments, we first pretrain the model on a synthetic 2-object dataset at for 60k iterations, as inputs, we sample 5-frame sequences with a 3-frame gap, each frame is of 128×128 resolution. We randomly mask out 3 out of the 5 frames during training, which we have explored the effects of varying these choices in Sec. 5.4.

During the second stage, we adapt our model to a test video by test-time training on the single video. We use the same frame sampling and masking strategy as the pretraining stage, with the exception that real videos are sampled with a frame gap of 1, which we empirically find yields better results. We implement all of our experiments in JAX and train on a TPU v3 with 16 cores. We refer the reader to supplementary material for additional details.

## 5.2 Evaluation

At inference time, we provide the model with the same number of frames as during training. We always mask out the last 3 frames in the sequence, and keep the middle frame prediction, and run the model over the entire video in a sliding window fashion. For numerical evaluation, we report IoU on binarized masks of each object layer. We render each object individually and use the resulting frames as the ground-truth alpha layer. To obtain the ground-truth binary mask $B_t^i$ for object $i$ at time $t$, we do a background subtraction, *i.e.* compute the squared distance between the ground-truth single-object scene and background layer, and threshold it: $B_t^i = \begin{cases} 1, & \text{if } \sum_C (Comp(\mathcal{L}_t^i, o_t) - \mathcal{L}_t^0)^2 \geq \beta \\ 0, & \text{otherwise} \end{cases}$ where $C$ indicates the RGB channels. We obtain the predicted binary mask using the same procedure.

## 5.3 Synthetic results and generalization to more objects

We train separate models on the 2-object Shadows and Reflections datasets and show results on both the 2-object and 4-object test sets in Table 1. For pretraining we use the estimated background (see supplementary for details), and for test-time training, we report both results from using the estimate (Ours-B) and from using the ground truth background (Ours-C). Using a more accurate background results in improved performance, but our model obtains strong results even with the estimated background. Despite only seeing 2-object videos during training, the model is able to generalize to scenes containing 4 objects due to it being structured to predict each layer separately, without knowledge of the other objects in the scene. Even without the test-time training stage, our data-driven method outperforms the single-video optimization method of Lu *et al.* [12] by a large margin. Finetuning on the test videos brings about further improvements in as few as 50 training steps, with additional performance gains as training time increases (see Figure 4).

Table 1: IoU for 2- and 4-object test sets. For 2-object videos, we evaluate on a random subset of 50 videos. 'TTT' refers to the number of test-time training iterations. 'Masks' refers to using the input object masks directly.

Figure 4: Performance (IoU) vs. number of test-time training iterations (TTT) for 4-object Shadows and Reflections test sets, using the ground truth background.

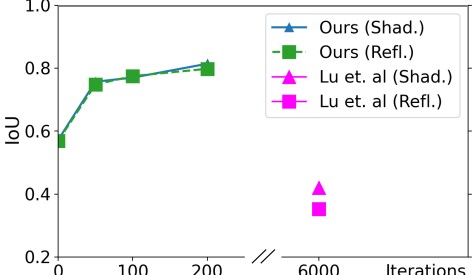

| Model | TTT it. | Shadows 2 ob ↑ | Shadows 4 ob ↑ | Reflections 2 ob ↑ | Reflections 4 ob ↑ |
|---|---|---|---|---|---|
| Masks | – | 0.372 | 0.368 | 0.348 | 0.317 |
| Lu et al. [12] | – | 0.469 | 0.420 | 0.451 | 0.353 |
| Ours-A | 0 | 0.607 | 0.572 | 0.593 | 0.569 |
| Ours-B (est. bg) | 200 | 0.802 | 0.752 | 0.806 | 0.747 |
| Ours-C (gt bg) | 200 | **0.843** | **0.814** | **0.843** | **0.798** |

We show qualitative results on synthetic videos from the Reflections dataset in Figure 5. The Omnimatte method of Lu *et al.* [12] struggles to correctly associate objects with their reflections,

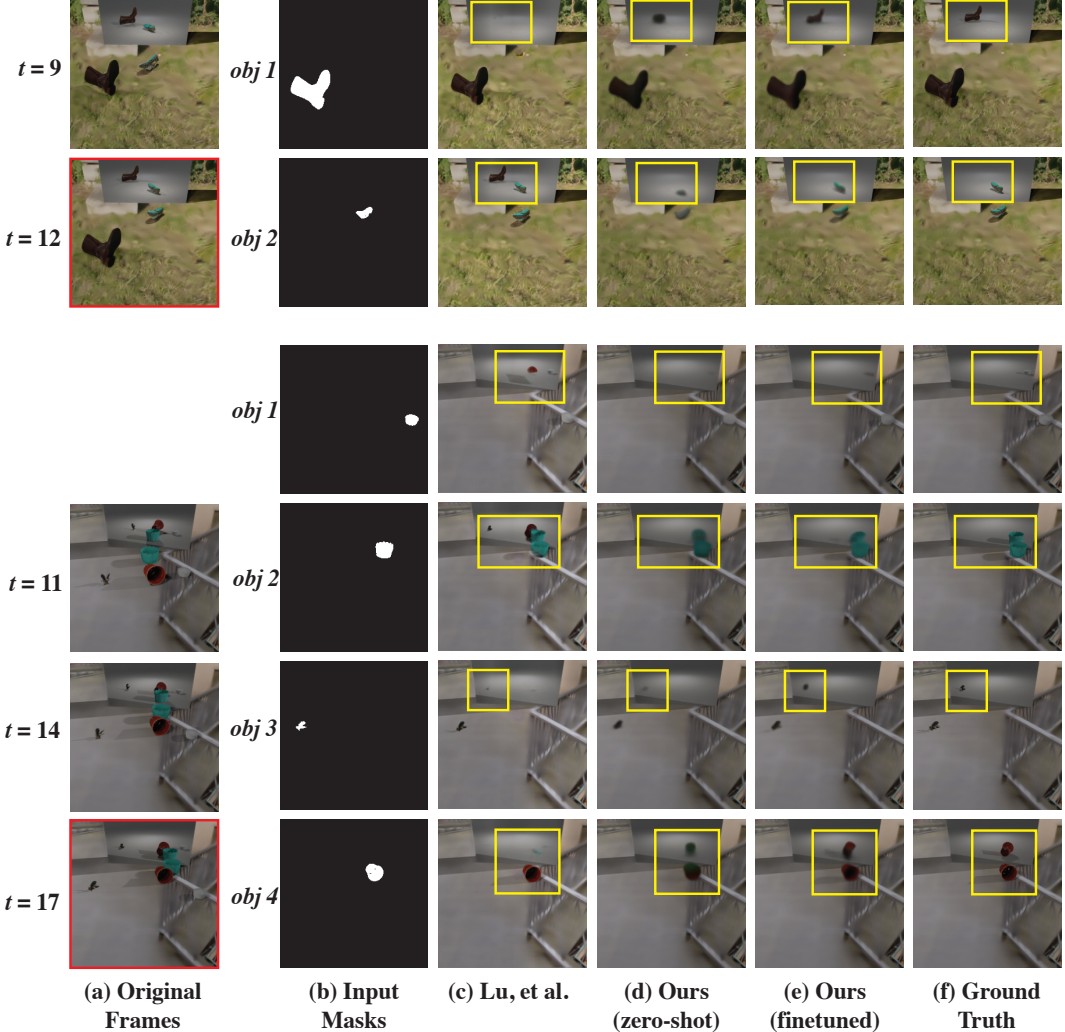

|  |  | (a) Original Frames | (b) Input Masks | (c) Lu, et al. | (d) Ours (zero-shot) | (e) Ours (finetuned) | (f) Ground Truth |

Figure 5: Results of training our model on 2-object scenes and testing on 2- and 4-object scenes. We show (a) the original video frames input to our model and the masked frame (red border) that must be reconstructed by our model, (b) the input object masks for the masked frame, (c) the result from [12], (d) our zero-shot result (pretrained network without test-time training), (e) our result after test-time training, and (f) the ground truth rendering. Whereas [12] places certain reflections in the incorrect object layer (yellow boxes), our pretrained network makes an accurate coarse association while test-time training further improves the reconstruction quality. Best viewed on a display, zoomed-in.

often assigning elements based on proximity (Figure 5, rows 1 and 2, column (c)). Our method, however, coarsely places the reflections in the correct layer out of the box (column (d)), reconstruction quality is further refined by test-time training, capturing additional details such as the shadows of the reflections (last row, column (e)).

The Omnimatte method [12] is a randomly initialized network that is optimized on a single video, and thus shorter sequences such as these provide a greater challenge due to the ambiguity (e.g. the motions of objects can appear correlated under a short observation window). Pretraining on many such short sequences, however, allows our method to overcome this limitation, and to do so in much fewer training iterations on the test video than is required for [12].

## 5.4 Ablations

We investigate the effects of pretraining with different numbers of masked input frames and different masking strategies. For all experiments, we report IoU on the 2-object datasets. In Table 2, we

sample 5 frame sequences during training and vary the number of frames $K$ that we mask. For these experiments we mask either the first or the last $K$ frames in the sequence. For $K = 0$, we apply the reconstruction loss to all of the frames, whereas when $K > 0$, reconstruction loss is applied only to masked frames. In Table 3, we set $K = 3$ and vary the method of selecting the masked frames: (i) random, (ii) inpainting (masking the middle frames), (iii) 'prediction' (masking the last or the first frames). For models trained under the random or prediction schemes, during inference time we generate masks under the prediction scheme and use the model output for the first masked frame only (i.e. frame index $5 - K$), running in a sliding-window fashion. As seen in Table 3, the choice of masking strategy does not have a significant effect. The number of masked frames similarly does not have a strong impact, as long as at least one frame is masked. When $K = 0$, performance drops significantly, showing the importance of the frame prediction task.

Table 2: Varying the number of input frames masked during training (out of 5). We report IoU and standard deviation computed from 3 random seeds.

| # Masked | Shadows ↑ | Reflections ↑ |
|---|---|---|
| 0/5 | $0.453 \pm 2\text{e-}4$ | $0.440 \pm 0.002$ |
| 1/5 | $0.679 \pm 0.012$ | $0.605 \pm 0.007$ |
| 3/5 | $0.638 \pm 0.002$ | $0.625 \pm 7\text{e-}4$ |
| 4/5 | $0.626 \pm 0.003$ | $0.617 \pm 0.004$ |

Table 3: Varying the strategy for masking the frame. We report IoU and standard deviation computed from 3 random seeds.

| Strategy | Shadows ↑ | Reflections ↑ |
|---|---|---|
| Random | $0.599 \pm 0.005$ | $0.611 \pm 0.004$ |
| Inpainting | $0.615 \pm 0.017$ | $0.605 \pm 0.004$ |
| Prediction | $0.585 \pm 0.003$ | $0.596 \pm 0.002$ |

## 5.5 Transfer to real videos

We adapt our pretrained model to real videos containing complex shadows, reflections, and trampoline deformations. We finetune our pretrained model for 1k iterations on each real video, taking about 12 minutes, compared with two hours for the optimization-based approach [12]. Our method achieves comparable results to single-video optimization (Fig. 6). The separation of effects in the `Trampoline` video is improved (rows 1 and 2), even though no trampoline effects are present in the training data, suggesting pretraining helps the network generalize to new effects. Other real videos are partially successful. In `Dogwalk`, the network separates the shadows of the dog and person correctly and even inpaints the shadow for most of the video (rows 3 and 4), but becomes confused as the dog obscures more of the person's shadow (rows 5 and 6). In this case, the single-video optimization method produces superior results, possibly due to our synthetic training data lacking large shadow occlusions. We created an additional synthetic dataset with larger shadows to better mimic these videos. Please see supplementary for additional results using this new dataset.

## 6 Discussion and Impact

We have demonstrated that a feed-forward network may be trained to successfully perform the Omnimatte task. On our synthetic dataset the feed-forward network even performs better than single-video optimization, showing the utility of the learned prior. Our model is trained with self-supervised losses, so it can be applied to real videos where ground-truth labels are not available, dramatically reducing the time required to produce video layer decompositions. Viewing the decomposition problem as a coordination game provides an intuitive explanation for the success of this approach: the network must learn to find a focal point of the game, and since each instance is tasked with predicting a single layer from a single object mask, the most prominent focal point lies where each instance predicts the effects of its object and nothing else.

**Ethical considerations.** Any technique that allows for new image editing effects could possibly be misused to produce fake or misleading images and videos. A layered decomposition of video only allows rearrangement of content already present, but even simple rearrangement can significantly alter the effect of a video. As with other research work in image editing, we hope publicly presenting the technique can help inform readers about editing capabilities that may exist in the near future.

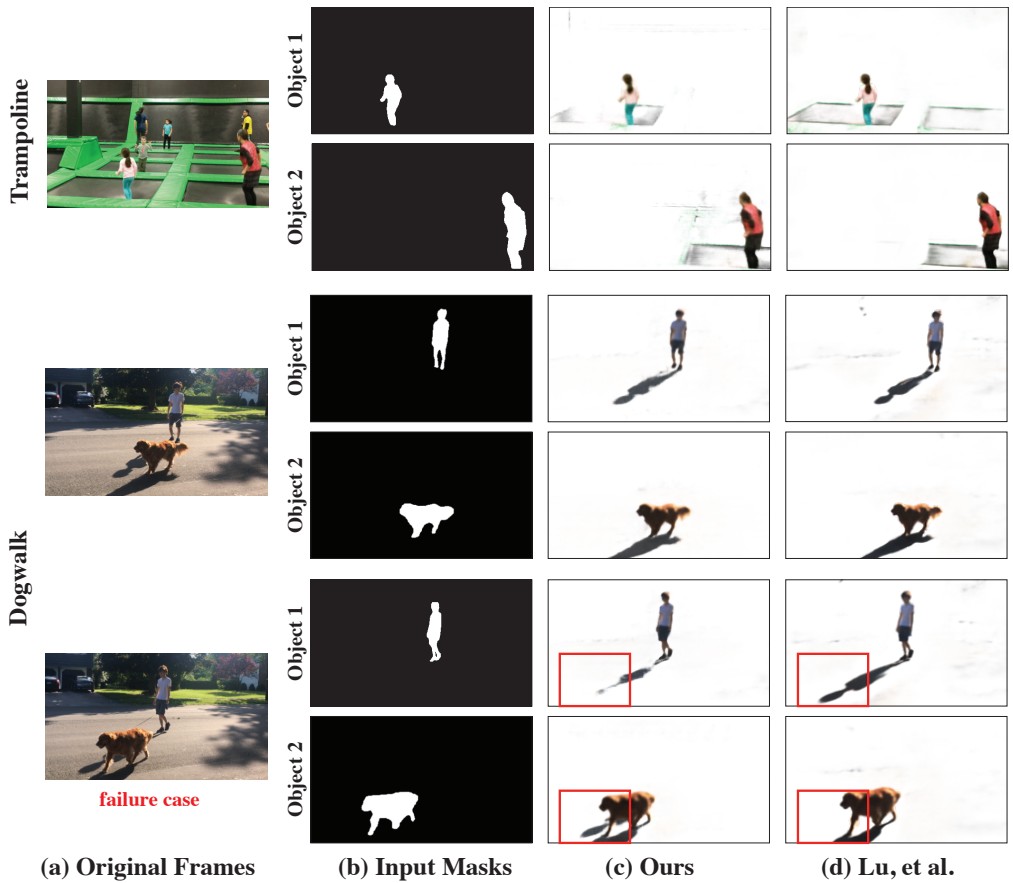

**(a) Original Frames**  **(b) Input Masks**  **(c) Ours**  **(d) Lu, et al.**

Figure 6: Results of our method on real videos. Our network pretrained on synthetic data requires much fewer training iterations (<10%) on test videos than Lu, *et al.*, and achieves comparable results in many cases (trampoline deformations and shadows grouped with the correct person/animal, top 4 rows). However, our method can fail in challenging scenarios involving heavy occlusion (part of the person's shadow is incorrectly placed in the dog's layer, rows 3 and 4).

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
