# Supplementary Material: Associating Objects and Their Effects in Video through Coordination Games

In this supplementary material, we provide additional implementation and training details, as well as results.

## 1 Model Architecture (Sec. 3.2)

**CNN Encoder.** We use the same architecture (based on pix2pix [1]) for our image encoder and our mask encoder: conv64-conv128-conv256-conv256, where each convN represents a block consisting of a $4 \times 4$ convolution with N filters and stride 2, followed by instance normalization [5] and leaky ReLU activation with negative slope 0.2. Following [3], we omit the instance normalization in the first convolutional block.

**CNN Decoder.** Our architecture consists of a series of convolutional transpose layers which perform 2x spatial upsampling: convt256-convt128-convt64-convt64-conv4, where convtN refers to a block consisting of a $4 \times 4$ convolutional transpose layer with N filters and stride 2, followed by instance normalization and ReLU activation. The final conv layer uses a $3 \times 3$ kernel with a tanh activation.

**Positional Embedding.** We use different learnable spatio-temporal positional embeddings for the queries (encoded masks) and keys (encoded images). The embedding is obtained by concatenating individual $x, y, t$ positional embeddings along the channel dimension, with individual channel sizes 96, 96, and 64 respectively, resulting in a final 256-dimensional embedding. We add these embeddings to the encoded masks and images before passing them to the transformer. During the test-time training stage, if the input video has larger dimensions than the training videos, we use bilinear interpolation to resize the positional embeddings.

**Transformer Decoder.** The transformer decoder consists of 4 heads and 2 layers. For an input video $I \in \mathbb{R}^{T \times H \times W \times 3}$ and a single object $i$, we obtain queries $Q^i = \mathcal{M}^i = \{m_1^i, \ldots, m_T^i\} = \{\Phi_{\text{mask}}(M_1^i), \ldots, \Phi_{\text{mask}}(M_T^i)\}$, keys $K = \mathcal{F} = \{f_1, \ldots, f_T\} = \{\Phi_{\text{enc}}(I_1), \ldots, \Phi_{\text{enc}}(I_T)\}$, and values $V = \mathcal{F}$; where $Q^i, K, V \in \mathbb{R}^{T \times \frac{H}{16} * \frac{W}{16} \times 256}$. After adding positional embeddings to $Q^i$ and $K$, we flatten $Q^i$, $K$, and $V$ to dimensions $T * \frac{H}{16} * \frac{W}{16} \times 256$ and pass them to the transformer decoder, which produces outputs of the same dimension. We reshape these outputs to $T \times \frac{H}{16} * \frac{W}{16} \times 256$ feature maps, and pass them through the CNN decoder to get the final RGBA layer prediction $\mathcal{L}_t^i$ for object $i$ at frame $t$. This procedure is repeated for all objects in the frame. In practice, we need only reconstruct the masked frames.

## 2 Losses (Sec. 3.3)

We train with self-supervised losses only—specifically, we apply supervision on the final reconstructed frame, *not* the individual layers (ground truth is used for final numerical evaluation only).

In Equation 2, the reconstruction is obtained by $\text{Comp}(\mathcal{L}_t, o_t)$, which denotes standard back-to-front alpha compositing [4] of the $N$ RGBA layers $\mathcal{L}_t^i$ according to known order $o_t$ (see Algorithm 1).

36th Conference on Neural Information Processing Systems (NeurIPS 2022).

**Algorithm 1** Alpha Compositing of Layers

---

$recon_t \leftarrow \mathcal{L}_t^0$
$i \leftarrow 1$
**while** $i < N$ **do**
    $recon_t \leftarrow \alpha_t^i \mathcal{L}_t^i + (1 - \alpha_t^i)recon_t$
**end while**

---

For the alpha sparsity term in Equation 3 in the main paper draft, we use $\gamma = 1$.

In Equation 5 in the main paper draft, our weighting coefficients are $\lambda_r = 0.05$, $\lambda_m = 1$ during the pretraining stage, and $\lambda_m = 0.1$ during the test-time training stage. We update $\lambda_m \leftarrow 0.1\lambda_m$ during the pretraining stage at 1k and 2k iterations. During the test-time training stage, we apply the same update at $0.1N$ and $0.2N$ iterations, where $N$ is the total number of training iterations.

## 3   Training (Sec. 5.1)

We use the Adamw optimizer [2] with an initial learning rate of 1e-3 for pretraining and 5e-4 for test-time training. The batchsize is set to be 128 for pretraining and 32 for test-time training.

## 4   Background Estimation (Sec. 5.3)

We estimate the static background $L^0$ for a stabilized video by computing the median value per RGB channel over video frames amongst valid pixels, which we define as those not falling within an object mask, i.e. $\sum_i M_t^i = 0$.

## 5   Additional Datasets (Sec. 5.5)

We experiment with creating a synthetic dataset which shares more visual similarities to our target real videos (e.g. Figure 6, main paper)—specifically, longer shadows, and more similar camera angles. We find that improving the realism of the synthetic data improves fine-tuning results on real videos (Figure 1).

This more realistic data proves more challenging than our initial synthetic dataset, with our zero-shot result often failing to capture effects (Figure 2), though many of these are subsequently captured during fine-tuning.

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

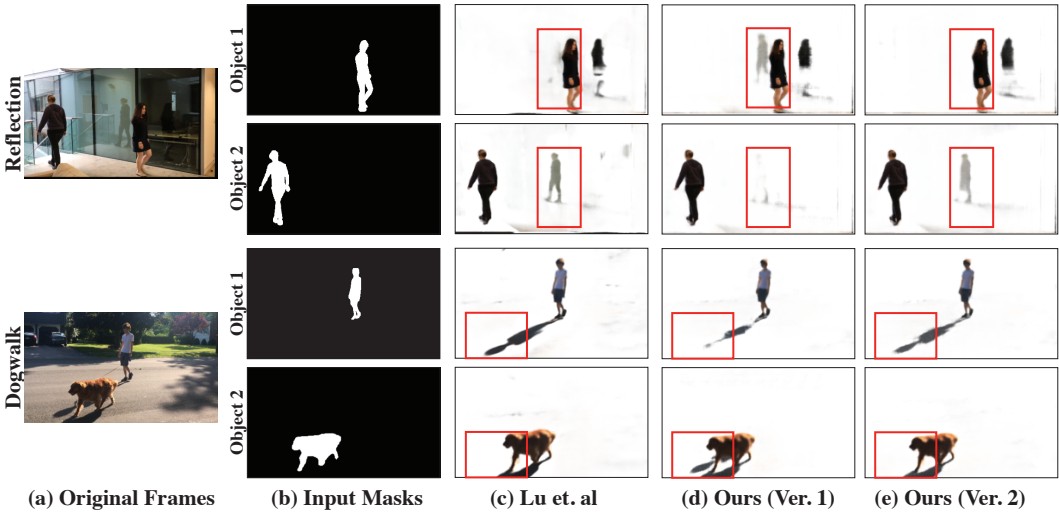

|  | (a) Original Frames | (b) Input Masks | (c) Lu et. al | (d) Ours (Ver. 1) | (e) Ours (Ver. 2) |

Figure 1: Results of our method after pretraining on various synthetic datasets and fine-tuning on real videos. We show (a), the original input frames; (b), the corresponding masks for each object; (c), the result from Lu, et al. [3]; (d), our initial result from pretraining on version 1 of our synthetic data; (e), our improved result after pretraining on version 2 of our synthetic data. (d) incorrectly places Object 2's reflection in Object 1's layer ("Reflection"), and part of Object 1's shadow in Object 2's layer ("Dogwalk"). (e) produces the correct associations and is on par with Lu, et al.

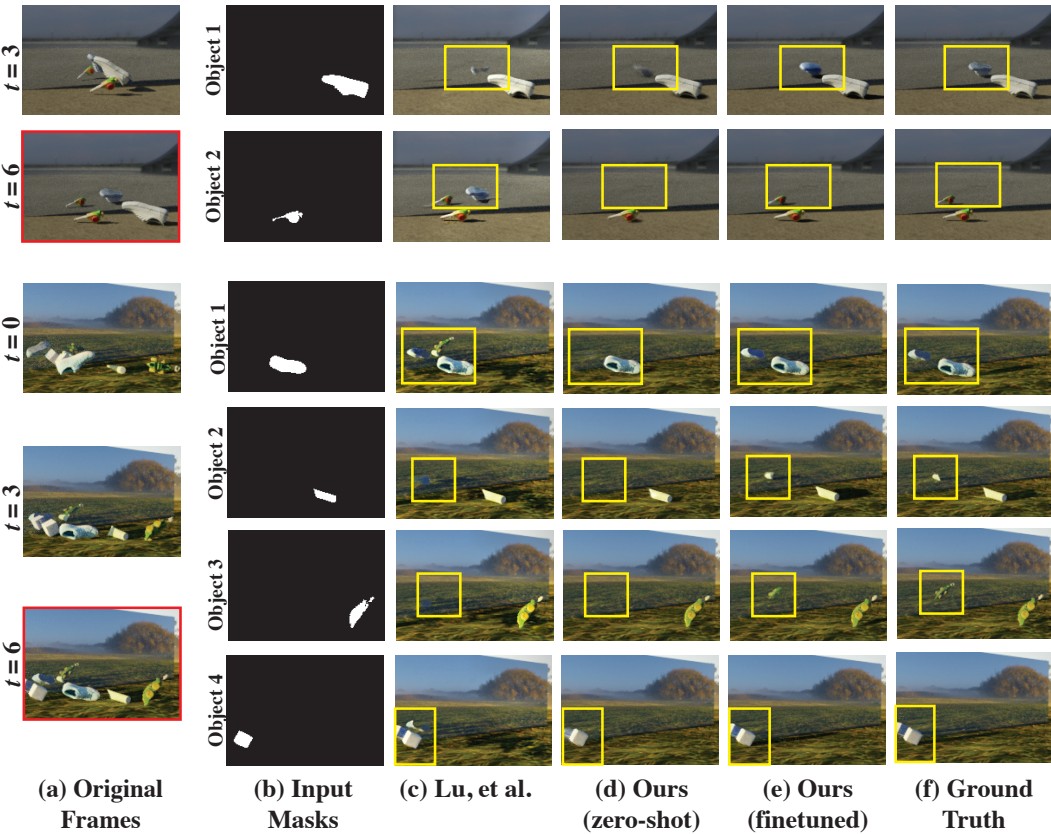

| (a) Original Frames | (b) Input Masks | (c) Lu, et al. | (d) Ours (zero-shot) | (e) Ours (finetuned) | (f) Ground Truth |
|---|---|---|---|---|---|

Figure 2: Results of our method on version 2 of our synthetic dataset. We show (a), the original video frames input to our model and the masked frame (red border) that must be reconstructed by our model; (b), the input object masks for the masked frame; (c), results from Lu, et al. [3]; (d), our zero-shot results; (e), our results after test-time training; (f), the ground truth rendering. Lu, et al. incorrectly groups reflections with the incorrect object (column (c), rows 2, 3, 6). Our zero-shot result fails to reconstruct some reflections (column (d), rows 2-5), but our fine-tuned result manages to capture all of the reflections in the correct layers (column (e)).