# OpenReview forum: "Associating Objects and Their Effects in Video through Coordination Games"
_NeurIPS.cc/2022/Conference — NeurIPS 2022 Accept_

### Official Review · Reviewer_7Czt · 2022-07-04

**Rating:** 7
**Confidence:** 3
**Soundness:** 3 good
**Presentation:** 4 excellent
**Contribution:** 3 good

**Summary:**

Given a short video with some moving objects and a rough mask for each object:
This paper tackles the problem of generating a per-object color and alpha mask for each object, containing all the effects on the image caused by that object (including e.g. shadows and reflections)
This paper achieves this via a network which plays a 'coordination game'; each 'copy' of the network is supplied with a different object's input mask, and attempts to reconstruct the mask and pixels corresponding to this object. The network is trained via a self-supervised reconstruction loss.

**Questions:**

Line 158: 2*sigmoid(5x); it seems there are two hyperparameters here, 2 and 5. Could the authors explain how these came about?

How sensitive overall is the network to the design choices? What would happen e.g. if any part of the overall loss was removed or hyperparameters changed?

**Limitations:**

Yes I think they have.

**Strengths And Weaknesses:**

## Originality

The problem being tackled is not original, but the proposed solution is, to the best of my knowledge, novel and interesting.

## Quality
Really nice to see experiments with real data rather than just rely fully on synthetic experiments.

I would have liked to have seen more 'simple, heuristic-driven' baselines. For example:
Given the videos are from a fixed camera, I could imagine using simple median-differencing to find a background image for each video.
Then this background image could be used to find per-frame, per-pixel differences from the background image; these must mostly be due to effects of foreground objects.
Finally each of these pixels can be associated to the foreground masks via e.g. nearest neighbour assignment.
I don't expect this simple heuristic to beat the proposed approach, but it might give better context to the numbers.

I would also have liked to see more ablations on the components of the algorithm. For example, what would happen if $W_t$ was set to ones everywhere? What about varying the hyperparameters in line 158: 2*sigmoid(5x)?

## Clarity

The overall writing of the paper is clear and easy to follow.

I enjoyed section 3.5: an explanatory view of the model being used.

I was very pleased to see the video results, I felt these made the overall system and quality of results clear.

I would have liked a little more justification of the formulation of $W_t$ (line 155). The idea of this is to give the network more emphasis on reconstructing areas *outside* the mask of the current object?


## Significance

The problem tackled is an interesting one, and the authors propose a thought-provoking solution.
I hope that the paper shows other researchers that this solution (of multiple identical networks playing a coordination game) can be of use in these types of scenarios.

---

> ### Author Response · Authors · 2022-08-02
> **Response to R4**
>
> Thank you for the comments and suggestions.
>
> **Wt (line 155)**
>
> The purpose of the spatial weighting term Wt is to encourage the network to focus on reconstructing the effects (shadows and reflections) in the output layers rather than focusing on accurately reconstructing the detail in the object appearance. Object appearance can be simply copied from the original video frame, as the object region is known and given by the input segmentation mask. If Wt is set to all 1s, more training steps are required before the effects begin to appear in the output layers, as the network focuses on reconstructing the object region.
>
> **Line 158 hyperparameters**
>
> These are the same hyperparameters used in the Omnimatte work for an approximate L0 loss, which allows the predicted alpha values to be closer to 1 to prevent solid objects from appearing semi-transparent.
>
> **Additional loss ablations**
>
> We use the same loss formulations as [12] and modify only the hyperparameter gamma in Equation 3 (which balances between the L0 and L1 terms in the alpha sparsity loss), and the scheduling of loss weighting coefficient lambda_m. We will ablate some of these components and add these results to the supplementary material.

---

> > ### Comment · Reviewer_7Czt · 2022-08-04
> > **Thank you**
> >
> > Many thanks for your replies here.
> >
> > I think these answers should definitely be incorporated into the text of your paper.

---

> > > ### Author Response · Authors · 2022-08-09
> > > **Thank you**
> > >
> > > Thank you for your response. We will incorporate this information into the revised version.

---

### Official Review · Reviewer_4zWN · 2022-07-09

**Rating:** 7
**Confidence:** 4
**Soundness:** 3 good
**Presentation:** 3 good
**Contribution:** 3 good

**Summary:**

This paper presents a novel frame for video layer decomposition, where they borrow the idea from game theory concept of focal points to frame this problem as a coordination game and let the networks reach consensus on their predictions.

**Questions:**

I wonder how this method deals with occluded scenes or objects that might be combined together or split up.

**Limitations:**

The presented method might fail with more complex scene with heavy occlusion.

**Strengths And Weaknesses:**

Strengths:

- The presented idea is both novel and interesting.

- The paper is well written and easy to read.

- Extensive experiments are conducted and improved results are shown.

Weakness:

- The results on real dataset is a little bit worse than the one on synthetic data.

---

> ### Author Response · Authors · 2022-08-02
> **Response to R3**
>
> Thank you for the questions and comments.
>
> **Occlusions**
>
> In this work we do not apply any special handling for occlusions between objects. If the segmentation mask of one object occludes another, the resulting layer will also have an occlusion. To address this issue, we could adopt an approach similar to [11] and inpaint the masks before passing them to our network, which would allow the network to inpaint the resulting layer. We leave this improvement to future work.

---

### Official Review · Reviewer_c3fb · 2022-07-10

**Rating:** 6
**Confidence:** 3
**Soundness:** 3 good
**Presentation:** 3 good
**Contribution:** 3 good

**Summary:**

The goal of this work is to decompose videos into different layers, for example, objects of interest and their shadow, reflection, and other visual effects. It is a challenging problem due to the complexity of the 3D geometry and lighting conditions in the real world, as well as the difficulties to get the ground truth. This paper proposes a self-supervised method to solve this problem. They borrow the idea from game theory and train networks to achieve this focal point. The experiments show the effectiveness of their design choice.

**Questions:**

Please refer to the weakness

**Ethics Review Area:**

["I don’t know"]

**Limitations:**

Please refer to the weakness

**Strengths And Weaknesses:**

Strengths:
+ The idea of using Focal Point is interesting and reasonable.
+ This method achieves promising visualization results for video decomposition.
+ The paper is easy to follow and the method is demonstrated detailedly.
Weakness:
- Object number: In this paper, the authors only show the results of 2 or 4 object scenarios. I am not sure this method can handle the scenarios with arbitrary objects. It might limit the generalization ability of this method.
- It is better to show how this network achieves this focal point.

---

> ### Author Response · Authors · 2022-08-02
> **Response to R2**
>
> Thank you for the comments and suggestions.
>
> **Object number**
>
> Our architecture processes object layers independently, and can thus generalize to different numbers of objects than were seen during training. All of the results on 4-object scenes in the paper and supplementary videos were from models trained on only 2-object scenes, demonstrating our method’s ability to generalize to different numbers of objects. Table 1 in Sec. 5.3 shows that the 2-object trained model performs well on 4-object scenes, with a smaller drop in IoU than [12].

---

### Official Review · Reviewer_Gkjr · 2022-07-10

**Rating:** 4
**Confidence:** 4
**Soundness:** 3 good
**Presentation:** 4 excellent
**Contribution:** 2 fair

**Summary:**

Authors propose a method that takes a video with multiple foreground objects as input, along with the corresponding background frames/image, and rough segmentation masks for each object. It then outputs an alpha decomposition of the video, where one layer corresponds to background, and the others to all visual effects from each object – so e.g. one layer includes a person plus their shadows and reflections. The method is trained for the task of missing-frame reconstruction, and relies on inductive biases (ease of learning) to ensure the correct shadow/etc. are paired with their respective objects, rather than any sophisticated constraints. Synthetic data (with plenty of shadows and reflections) is used for training. At test time, the method is further tuned to optimise the decomposition of a given video. It is demonstrated on both synthetic and real data; quantitative results on synthetic data are better than a recent baseline, and qualitative results on real data look reasonable.


**Questions:**

See issues raised above under 'weaknesses'.

How sensitive is the method to the accuracy of the provided object segmentations? If it can recover from imperfect masks and still capture the complete object, this might also be of interest in itself.

It would be interesting to see a more detailed analysis of failure cases. For example, bin the data according to how close the relevant 'effect' is to the object itself, and see how accuracy varies with distance – or see how problematic proximity/overlap with other objects (as in the Dogwalk failure) is in general

Why are the losses denoted mathbf capital E, when in fact they're just scalars?


**Limitations:**

There is minimal discussion of limitations; the paper would benefit from adding an explicit subsection for this. There is adequate discussion of broader impacts.

**Strengths And Weaknesses:**

Strengths:

- the proposed approach is novel

- the proposed method achieves significantly better quantitative results on synthetic data than a recent baseline

- the method is also demonstrated on real-world data, where it often achieves visually acceptable results

- the paper is clear and easy to read throughout; it is well structured, and the figures are appropriate

Weaknesses:

- the assumption of known background seems rather restrictive (and makes the problem considerably easier)

- given the background is assumed known a-priori, what is the actual practical purpose/use of the proposed method? Authors should identify this clearly in the introduction, and add an experimental evaluation that shows performance on this task

- on real data, the method is not significantly better than the baseline (though it is faster if one disregards training time)

- there is no attempt to measure quantitative performance on real-world data. While I appreciate that this would require some manual annotation, it would only need to be a handful of frames from the few videos, indicating which regions are indeed moving with which object

- the technical contribution is small, compared with the baseline [12], and also considering similar-in-spirit works such as Video Centrifuge – particularly given the method uses fairly standard architectures and doesn't have a strong justification for its own successes (see below)

- there is no convincing reason presented for the method to work – the 'correct' result is one of several equivalent local optima (assigning shadows to arbitrary layers), and the arguments about Schelling points do not resolve why the 'correct' result is found (merely why the layers should find some arbitrary valid joint decomposition). It seems that the correct optimum is found simply because it is 'easier' for the network to learn, but it'd be nice to see a proper analysis of this. At minimum, does the model training ever converge to 'incorrect' solutions – and for what fraction of training runs if so?

- the method assumes (I think) data is in sRGB (normalised to 0–1) and that different object contributions can be combined by alpha blending. However, this is not true in general – reflections should be treated as strictly additive in linear color space, and shadows darken surfaces rather than alpha-overlaying. This may limit applicability to scenes where the lighting and exposure are fairly well-behaved

- resolution is limited (only 128x128 even training/testing on TPUs), thus limiting applicability in practice

---

> ### Author Response · Authors · 2022-08-02
> **Response to R1**
>
> Thank you for the detailed comments.
>
> **Technical contribution**
>
> The Visual Centrifuge work trains a network to decompose an input video in a fully supervised manner, where training data is created by alpha blending two random videos from a dataset. Unlike our work, the layers cannot be conditioned on an input object mask, and VC is not designed to separate each object and its effects into separate layers. Due to its architecture, VC is also unable to generalize to more layers than seen during training, while our method can (Section 5.3, where we train on 2 objects and test on 4 objects).
>
> As mentioned previously, our method produces more consistent results across random initializations than Omnimatte. For the Reflection example, it yielded a correct result 4/4 times vs. Omnimatte’s 2/4 [[link to visualization](https://drive.google.com/file/d/13YeLLTQBM0prwb3AxLL2RuRvjyzOtnAt/view?usp=sharing)].
>
> **Known background, applications**
>
> The learned model takes as input a background image. In practice, for a given video, we estimate that background image by registering the video with homographies (as in the Omnimatte work [12]) to produce a camera-stabilized version, then compute a global median background that can be mapped back to the original frames using the computed homographies.
>
> The goal of our work is to associate effects with the correct moving object when multiple objects exist in the scene. Even with a ground-truth foreground / background separation, separating the foreground into objects and associated effects remains a challenging problem (that doesn’t specify which effect belongs to which object). Applications of this separation could include removing distractor objects from videos where multiple objects are present, retiming individual people in video (as in [11]), or creative editing effects like stroboscopic videos (as in [12]).
>
> **Limited resolution**
>
> The method is not limited to 128x128 resolution; we chose this size as it allows for shorter experiment runtimes (3 hours for pretraining).
>
> **Correctness of alpha blending**
>
> It is true that the effects we model (principally shadows and reflections) are not equivalent to alpha blending. Nonetheless, alpha blending has been shown previously to be an effective approximation (VC [1], “Obstruction-Free Photography”, Xue, et al., 2015), and allows us to approximate reflections, shadows, and occlusions (e.g., Dogwalk) with a single image formation model. A linear image formation model introduces difficulties in modeling common cases such as occlusions and overlapping shadows. Improving our formation model to capture additive and subtractive effects is an interesting direction for future work.
>
> **Segmentation mask accuracy**
>
> On synthetic data we have results for eroding the input object masks and generating the complete object. We will add these to the supplementary. The finetuning on real data uses estimated masks from off-the-shelf tools such as Mask RCNN, indicating that the method can recover from imperfect masks to a certain extent.

---

### Author Response · Authors · 2022-08-02
**Response to Shared Questions**

We thank all the reviewers for their constructive comments. Here, we address shared questions.

**Why is the focal point the correct solution? (R1, R2)**

The optimization must find a focal point in order to satisfy the loss, but as R1 points out, the focal point does not necessarily lie at the correct solution. In fact, the baseline method without masked frames (Table 2, “0/5 Masked”) does have a focal point at an _incorrect_ solution: the layer separation is worse than configurations with masking, even though the reconstruction error (training loss) is better (see column (g) [here](https://drive.google.com/file/d/1kY5pEvng4cfVwC0l83ychtigFxOVrICO/view?usp=sharing)). To push the focal point to the correct solution, we withhold one or more frames (Table 2, “x/5 Masked”) from the network. In this masked configuration, we observed that the optimization found the correct solution (focal point) in all of our synthetic data experiments (see convergence stability, below). We will amend the text and include total reconstruction error in Table 2 to highlight this effect.

**Convergence stability (R1, R4)**

The pretraining stage always converges to the correct result in our experiments, with small variation in error over runs (Table 2, std. dev).

When test-time finetuning on real videos, the model may occasionally converge to an incorrect solution, however we still notice much more stable behavior compared to Lu et al. [12]. For example, we pretrained 4 different models using different seeds, and then finetuned each model on the Reflection example, and obtained 4/4 correct decompositions as compared with Lu et al.’s 2/4 [[link to visualization](https://drive.google.com/file/d/13YeLLTQBM0prwb3AxLL2RuRvjyzOtnAt/view?usp=sharing)]. See Lu, et al. [12] Sect. 4.8 for a discussion of stability issues with that method.

---

### Meta-Review · Area_Chair_mJF2 · 2022-08-26

**Recommendation:** Accept
**Confidence:** Less certain

**Metareview:**

All reviewers found that the paper provides a novel, interesting solution, and is well written. They appreciated that the proposed method outperforms prior work on synthetic experiments and shows reasonable results on real data. The video results were particularly helpful in judging the results.

The majority of the reviewers were concerned about the convergence of the proposed coordination game to the correct solution. While the authors provided some empirical evidence, a more formal analysis could alleviate concerns much more easily and would provide a strong justification for the proposed method.

The requests by reviewers for more ablations, simple heuristic baselines, and quantitative results on real data were simply ignored by the authors. This does not induce confidence that any of these requests will be addressed in a final version.

**Award:**

No

---

### Decision · Program_Chairs · 2022-09-14

Accept